# Label-free nanoscale optical metrology on myelinated axons in vivo

Junhwan Kwon[1,2], Moonseok Kim[3], Hyejin Park[1,4], Bok-Man Kang[1,2], Yongjae Jo[1,2], Jae-Hwan Kim[1,2], Oliver James[1], Seok-Hyun Yun[3], Seong-Gi Kim[1,2], Minah Suh[1,2] & Myunghwan Choi [1,2]

In the mammalian nervous system, myelin provides electrical insulation for the neural circuit by forming a highly organized, multilayered thin film around the axon fibers. Here, we investigate the spectral reflectance from this subcellular nanostructure and devise a new label-free technique based on a spectroscopic analysis of reflected light, enabling nanoscale imaging of myelinated axons in their natural living state. Using this technique, we demonstrate three-dimensional mapping of the axon diameter and sensing of dynamic changes in the substructure of myelin at nanoscale. We further reveal the prevalence of axon bulging in the brain cortex in vivo after mild compressive trauma. Our novel tool opens new avenues of investigation by creating unprecedented access to the nanostructural dynamics of live myelinated axons in health and disease.

[1] Center for Neuroscience Imaging Research (CNIR), Institute for Basic Science (IBS), Suwon 16419, Republic of Korea. [2] Department of Biomedical Engineering, Sungkyunkwan University, Suwon 16419, Republic of Korea. [3] Harvard Medical School, Boston 02114 MA, USA. [4] Department of Biological Science, Sungkyunkwan University, Suwon 16419, Republic of Korea. Junhwan Kwon and Moonseok Kim contributed equally to this work. Correspondence and requests for materials should be addressed to M.C. (email: photomodulation@gmail.com)

In vertebrates, most axon fibers are enwrapped by myelin, a multilayered membranous sheath. The myelin serves as an electrical insulator for the axon to provide rapid and energy-efficient conduction of neural information[1, 2]. A subtle change in the myelin nanostructure can thus confer substantial changes in conduction speed and remodel the neural network[3]. Abnormal myelination caused by genetic mutations and physical injury is associated with various neurological disorders and mental illnesses[4]. Recent studies further suggest that myelination is dynamically regulated by intimate axon–myelin interactions[4], even under physiologic conditions, serving as a regulatory mechanism for neuronal plasticity, long-range synchrony, and network-level oscillation[5–8].

Given the critical role of myelin in normal and diseased states, the ability to monitor the nanostructural dynamics of live myelinated axons would be highly advantageous. However, none of the current imaging techniques can provide access to the nanoscale information of myelinated axons in their natural living environment. Electron microscopy has been a gold-standard technique in the study of myelin nanostructures because it offers atomic resolution, which permits the visualization of the compact multilayers of myelin at the nanoscale[9]. Recent advances in serial tomography have resulted in pioneering discoveries in the myelin field, such as heterogeneous coverage of myelin in cortical axons[10, 11]. However, electron microscopy is only applicable to fixed or frozen tissues and is therefore unsuitable for studying functional dynamics in living systems. By contrast, a multitude of optical techniques are applicable to living animals with minimal invasiveness[12]. Various contrast mechanisms for myelinated axons have been developed, including fluorescence with exogenous probes[13, 14], optical

coherence tomography[15], Raman scattering[16, 17], and third-harmonic generation[18]. These technical advances enabled observation of dynamic cellular processes in live myelinated axons in physiology and pathophysiology. In particular, label-free techniques, such as Raman scattering and third-harmonic generation, have high potential for clinical translation[18, 19]. However, these microscopic techniques have spatial resolutions greater than the optical diffraction limit of ~200 nm, rendering them inadequate for studying the nanostructures of interest in the myelinated axon (e.g., a cytosolic layer in myelin ≈3 nm). Even super-resolution techniques have insufficient resolution (>10 nm) for delineating myelin substructures and are worsened by scattering and aberrations in thick biological tissues[20, 21].

Recently, Schain et al.[22] showed that spectral reflectance from nervous tissues provides label-free, high-contrast imaging of myelinated axons in vivo. This technique named SCoRe (spectral confocal reflectance microscopy), utilized several discrete laser lines for generating reflectance images of myelinated axons and was shown to be highly useful for qualitative detection of myelin development and degeneration. Here, we report that the spectroscopic analysis of broadband light reflection from myelinated axons provides quantitative examination of the multilayered cytoarchitecutre at nanoscale precision. The physical principle of this technique is based on the spectral reflectometry (SpeRe), which is widely used in the state-of-the-art semiconductor industry to measure layer thicknesses of a multi-film stack with sub-nanometer accuracy[23, 24]. By adopting and optimizing this technique to mammalian axons, we demonstrate live measurements of nanoscale dynamics in myelinated axons in response to physiologic osmotic modulation and mechanical brain injury.

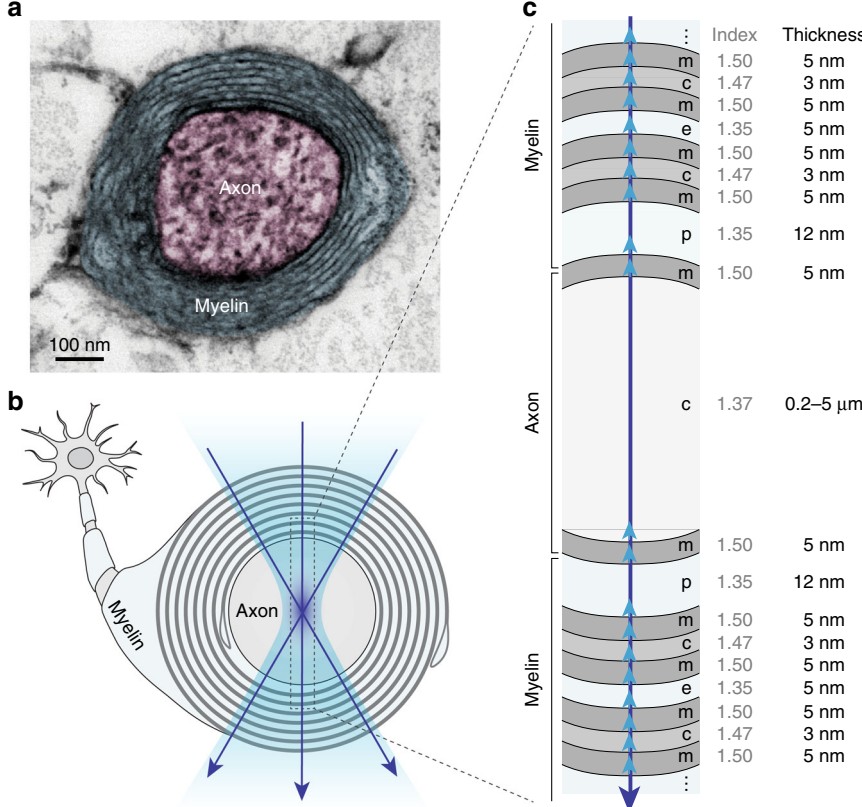

**Fig. 1** Thin-film model of the myelinated axon. **a** Cross-sectional image of a myelinated axon in the mouse brain obtained using transmission electron microscopy (TEM). **b** A schematic cross-sectional view of the myelinated axon, with an interrogating light beam (purple arrows) focused at the center of the axon. **c** Physiologic thin-film model for the myelinated axon with corresponding refractive indices and thicknesses indicated. Upward arrows (light blue) indicate Fresnel reflections at the interfaces. Not drawn to scale. m, membrane; c, cytosol; e, extracellular space; p, pericellular space

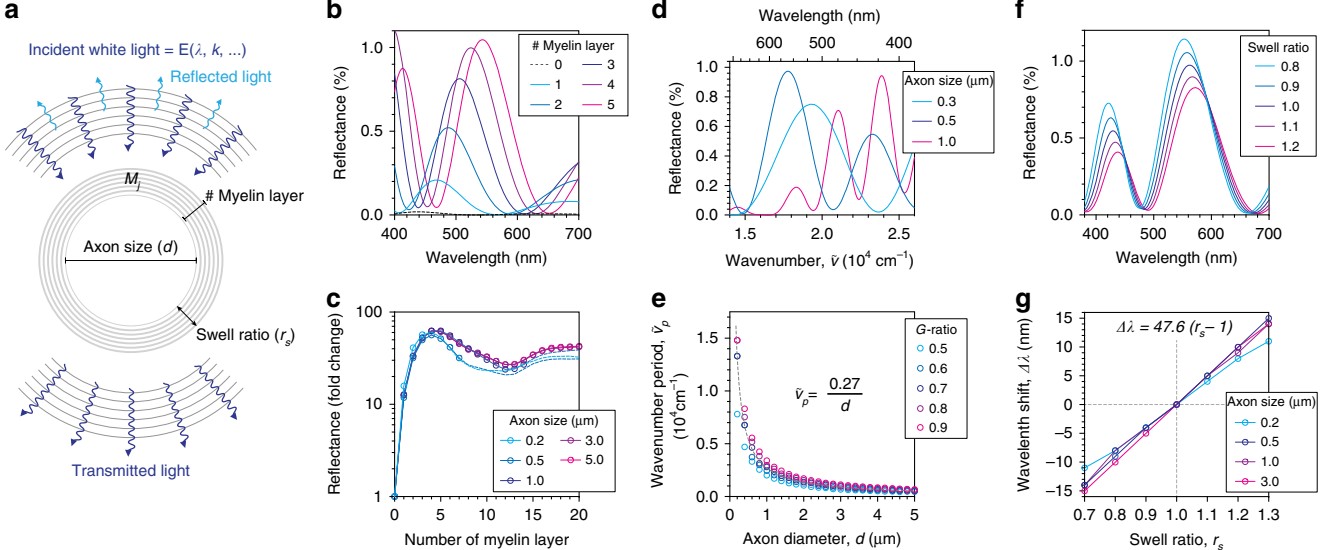

**Fig. 2** Theoretical background on SpeRe. **a** Scheme of optic simulation for SpeRe. Structural variables of interest in myelinated axons are annotated. Absolute reflectance (%) for each wavelength is simulated by applying the theory of electromagnetic waves. **b** Simulated reflectance spectra with variable myelin layers. The axon size is set to 0.5 µm. **c** Fold increase in visible reflectance (400–700 nm) relative to unmyelinated condition. **d** Simulated reflectance spectra with varying axon size. Horizontal axis shows the wavenumber, $\tilde{\nu}$ (inverse of wavelength). The g-ratio (i.e., the ratio of inner axonal diameter to the total outer diameter) is set to 0.7. **e** Relationship between wavenumber period $(\tilde{\nu}_p)$ and axon diameter (d). Dotted line indicates fitted curve to hyperbola ($R^2 = 0.99$), and the best-fit equation is shown. **f** Simulated reflectance spectra with a varying swelling ratio of the extracellular space between the myelin layers. The axon size and g-ratio at the isosmotic condition are set to 0.5 µm and 0.7 (i.e., 6 myelin layers), respectively. **g** The relationship between spectral shift (Δλ) and swelling ratio ($r_s$). The equation is the best fit to linear regression ($R^2 = 0.98$)

## Results

**Principle of SpeRe.** Myelin wraps around the axon in a compact, multilayered spiral (Fig. 1). In a cross-sectional view, it forms nearly concentric rings, alternating cellular space with a high refractive index ($n = 1.47$) and extracellular space with a lower refractive index ($n = 1.35$)[25–28]. When light is focused at the center of the axon, the incoming light wave is partially reflected off from the interface owing to discontinuity in the refractive index, as described by Fresnel's Law (Fig. 1b, c). The reflected waves from each interface have different optical path lengths, and they interfere with the relative phase shift, as determined by the physical thickness and the optical wavelength in each layer. This phenomenon, termed as thin-film interference, indicates that nanostructural information is spectrally encoded in the reflected light. SpeRe utilizes this principle to measure the thickness of thin film coatings on a planar substrate with nanometer precision[24]. Therefore, we reasoned that the reflectance spectrum from the myelinated axon may be decoded to obtain its nanostructural information.

To understand the quantitative relationship between the reflectance spectrum and the nanoscale cytoarchitecture, we performed numerical simulations based on the wave theory of light (Fig. 2a, Supplementary Notes 1 and 2). Simulation parameters were either obtained from literature or estimated based on reported values (Supplementary Table 1) and validated by transmission electron micrographs of myelinated axons from mouse brain and spinal cord (Supplementary Figs. 1 and 2). Interaction of light waves at the subcellular layers (i.e., Fresnel reflection) was described by the thin-film matrix theory, and distribution of light waves at the focus was formulated by the vector diffraction theory[29, 30].

Using the simulation, we first investigated how layers of myelin around a cylindrical axon affect the magnitude of reflected light in the visible region. Bare axons with no myelin layers (unmyelinated axon) reflect less than 0.1% of the input power. However, adding even a single myelin layer increased the visible

reflectance by over an order-of-magnitude (Fig. 2b, c). Regardless of the size of the axon, the reflectance increased with the number of myelin layers, but after 4–5 layers the reflectance became saturated with further increase in the number of layers (Fig. 2c). This behavior is likely due to phase mismatch between adjacent myelin layers having thin optical path length. This result indicates that the magnitude of reflectance alone may provide sufficient contrast to distinguish myelinated axons from unmyelinated ones. Next we studied how the axon diameter affects the spectrum of reflected light (Fig. 2d, e). We found that the reflectance spectrum is a periodic function of wavenumber, $\tilde{\nu}$ (inverse of wavelength). Interestingly, the wavenumber period $(\tilde{\nu}_p)$ is inversely proportional to axon diameter (Fig. 2e). Physiologic variation in the thickness of the myelin sheath (i.e., the g-ratio, the ratio of inner axonal diameter to the total outer diameter) only modestly contributed to this inverse relationship. By utilizing a regression analysis, we obtained the formula shown in Fig. 2e, which formed the basis for elucidating the axon diameter (d) from the reflectance spectrum. Last, we studied the effect of myelin swelling on the reflectance spectrum (Fig. 2f, g). Myelin swelling occurs predominantly in the extracellular space between myelin layers under physiologic variations in osmotic pressure[31, 32]. Simulations with varying thicknesses of the extracellular space indicated that myelin swelling led to blue shift in the reflectance spectrum. The degree of the spectral shift was linearly proportional to the swelling ratio and had little dependency on axon size (Fig. 2g). This result indicates that myelin swelling is quantifiable at the nanoscale by measuring spectral shift. Collectively, our simulation results suggest that SpeRe enables us to investigate subcellular structures of myelinated axons at the nanoscale, such as the degree of myelination, axon diameter, and myelin swelling.

**SpeRe on nervous tissues.** To measure the reflectance spectrum in myelinated axons in situ, we used a confocal microscope in back-reflectance mode with a spectrally scanned white-light laser

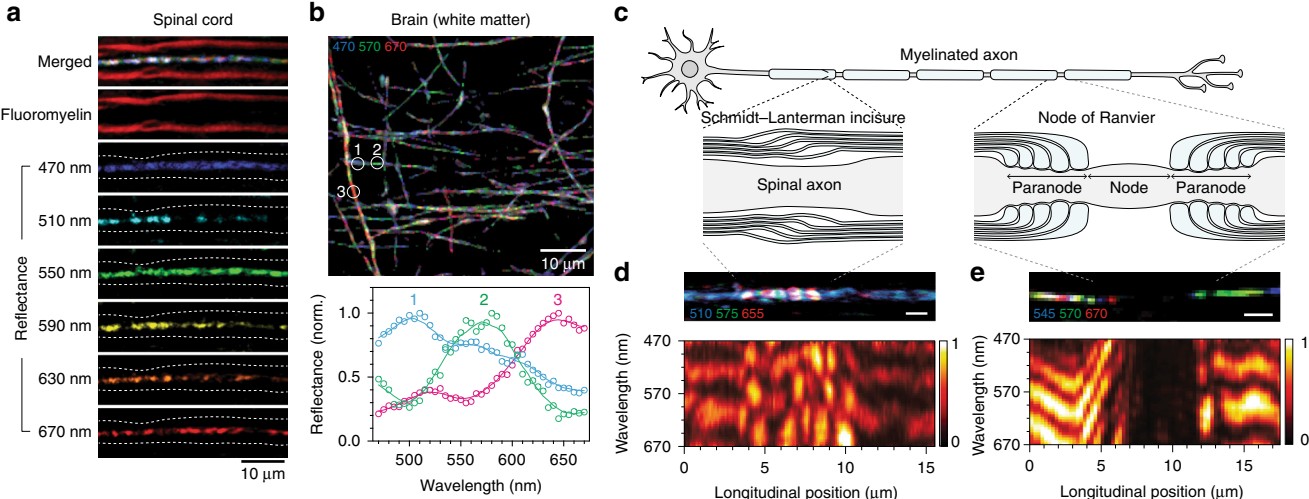

**Fig. 3** SpeRe on nervous tissues. **a** SpeRe images of the mouse spinal cord with fluorescent counterstaining of the myelin (fluoromyelin). Dashed lines in reflectance images indicate axon−myelin interfaces. Note that the reflectance signal is localized at the centerline. **b** A representative SpeRe image of the mouse brain near the corpus callosum (470 nm for blue, 570 nm for green, 670 nm for red) and representative reflectance spectra at the labeled axon segments. **c** A schematic illustration of the myelinated axon in a spinal neuron. **d** A reflectance image and spectral map at Schmidt–Lanterman incisure. Scale bar, 2 μm. **e** A reflectance image and spectral map at the node of Ranvier. Scale bar, 2 μm

as an input source (Supplementary Fig. 3). In agreement with our simulation and previous reports[22, 33], myelinated axons in nervous tissues exhibited a notable light reflectance, compared to that of the surrounding tissue in SpeRe images (Fig. 3a, b). Reflectance spectra from each axon segment were extractable at a high signal-to-background ratio (>10) for reliable quantitative analyses (Fig. 3b). To validate the source of the signal, we counterstained with a fluorescent probe selective to myelin (fluoromyelin)[34]. As shown in Schain et al., the reflectance signal was specifically observed only at myelinated segments and was located only at the centerline (Fig. 3a)[22]. This phenomenon can be simply explained by reflection geometry (Supplementary Fig. 4). When focused at the center of the axon, the incoming light rays are perpendicular to the reflection interfaces (i.e., perpendicular tangent theorem). The reflected rays thus follow the same path as the incoming light in a backward direction, satisfying confocality, and therefore they are detected through the confocal pinhole. When the focus is shifted away from the center, reflected light is tilted and mostly rejected by the pinhole. For this reason, the apparent diameter of a SpeRe image on axon is nearly diffraction-limited and independent of the axon size (Supplementary Fig. 5). Moreover, confocal reflection is limited to axons that are nearly orthogonal to the optical axis. The angular limit was measured to be approximately ±10° in our imaging setup (Supplementary Fig. 6).

It has also been reported that multiple label-free techniques, including spectral reflectance[22], coherent Raman scattering[17], and third-harmonic generation[18], can be used to identify specialized axonal structures, such as Schmidt-Lantermann incisure and node of Ranvier (Fig. 3c). We further studied if SpeRe on these specialized structures reveals more structural information[35]. To visualize spectral information along the longitude of the axon, we presented the data as a spectral map (xλ). In the spectral map, the incisure, a cone-shaped loosening of the myelin sheath, showed a characteristic speckled pattern (Fig. 3d, lower panel). This speckled spectral feature is conceivably due to abrupt longitudinal change at the loosening of each myelin layer. Additionally, interspersed unmyelinated segments (nodes of Ranvier) exhibited over an order-of-magnitude lower reflectance, as expected from the simulation (Fig. 3e and Supplementary Fig. 7). Reflectance at adjacent paranodes gradually increased and exhibited near maximal reflectance at the internode. In the spectral map, the paranode often showed a progressive spectral shift, conceivably due to the gradual structural change. These observations indicate that spectral features may offer a way to quantify geometric parameters of the axon, such as the length of the incisure, node of Ranvier, and internode, in a label-free manner.

**Mapping the axon caliber**. Our wave simulation suggested that the wavenumber period $(\tilde{\nu}_p)$ is an excellent estimator of the axon diameter (Fig. 2d, e). To test feasibility, we first confirmed that this metric $(\tilde{\nu}_p)$ precisely estimates diameters of synthetic microfibers (Supplementary Fig. 8 and Supplementary Movie 1). We further validated the SpeRe measurement on monodisperse synthetic microbeads by using scanning electron microscopy (Supplementary Fig. 9). We next applied this approach to nervous tissues ($n = 8$ for spinal cord, $n = 16$ for brain cortex) and compared the results with a conventional fluoromyelin-based method. As fluoromyelin often stains the outer rim of the myelin due to limited penetration into the myelin sheath, we estimated the axon diameter by assuming a g-ratio of 0.7, which is well known to be conserved in mature axons[34]. Owing to diffraction-limited resolution of the fluoromyelin-based method, small axons with indistinguishable outer rims ($d < 300$ nm) were excluded. For analysis of spectral data, we applied empirical mode decomposition (EMD) to reliably extract wavenumber periodicity (Online Methods). Surprisingly, axon calibers estimated using independent methods resulted in nearly identical values, demonstrating the validity of the SpeRe method in estimating the diameter of the axon (Fig. 4a and Supplementary Movie 2).

We subsequently reasoned that this structural information for each axon segment could be reconstructed into a nanoscopic axon map. As a proof of this principle, we skeletonized axon fibers from the image stack, quantified the axon caliber at each axon segment at ~1 μm intervals by spectral periodicity, and combined the structural information to reconstruct a comprehensive three-dimensional view (Fig. 4b and Supplementary Movie 3). In histogram analysis, this nanostructural information on axon caliber exhibited a characteristic log-normal distribution, with a geometric mean of 0.52 μm and a geometric standard deviation of 0.37 (Fig. 4c), which was consistent with previous findings by large-scale post-mortem electron microscopy[36].

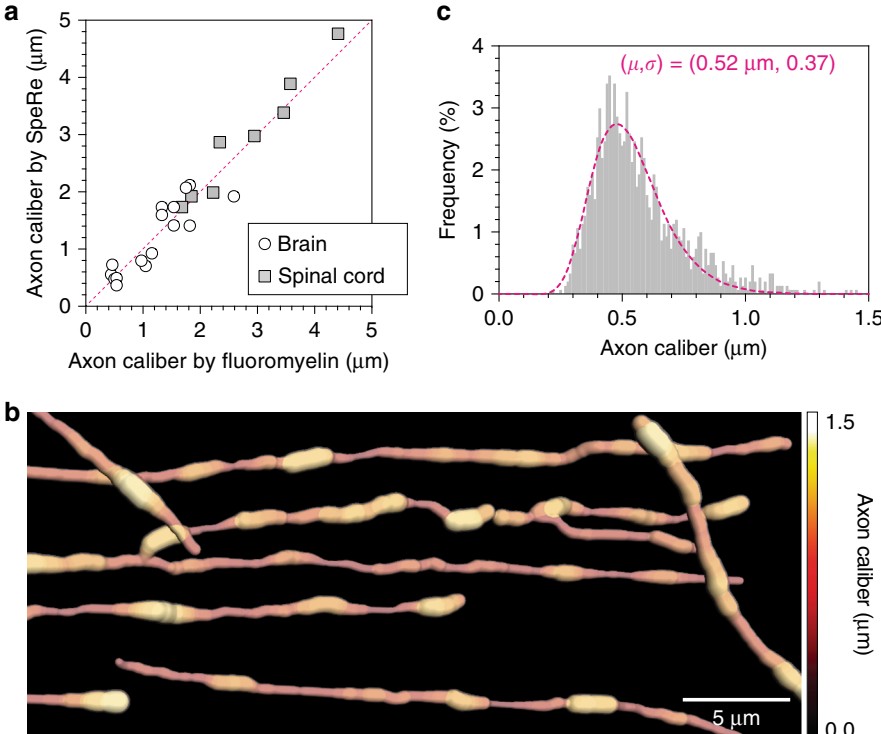

**Fig. 4** Nanoscale mapping of axon caliber. **a** Validation of axon caliber, as measured by SpeRe. Diameters of axons from the brain ($n = 16$) and the spinal cord ($n = 8$) were quantified using both SpeRe and a fluoromyelin-based method. The red dotted line indicates the unity line ($R^2 = 0.93$). **b** A nanoscale volumetric view of the axon. The image stack (xyzλ) is skeletonized and rendered using the axon diameter extracted from wavenumber period ($\tilde{\nu}_p$) for each pixel on the skeleton. Myelin is not shown. **c** Probability density function of axon diameter acquired from **b**. The diameter is quantified for each 1.2 μm axon segment. The dotted line indicates the best fit to log-normal distribution (geometric mean = 0.52 μm, geometric standard deviation = 0.37)

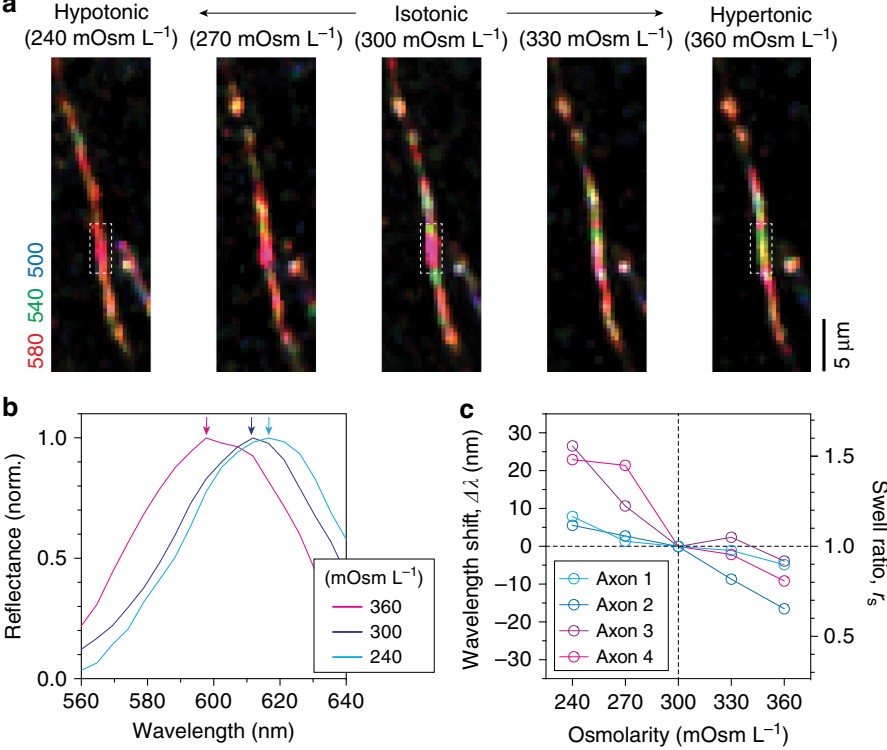

**Fig. 5** Nanoscale sensing of myelin swelling under osmotic challenge. **a** SpeRe on the myelinated axon in the freshly sliced brain cortex under osmotic challenge. The osmotic pressure changed from 240 to 360 mOsm L$^{-1}$. **b** Representative reflectance spectra from the demarcated regions in **a**. Arrows in corresponding colors indicate the peaks. **c** Quantification of spectral shift in response to changes in osmotic pressure ($n = 4$ axons)

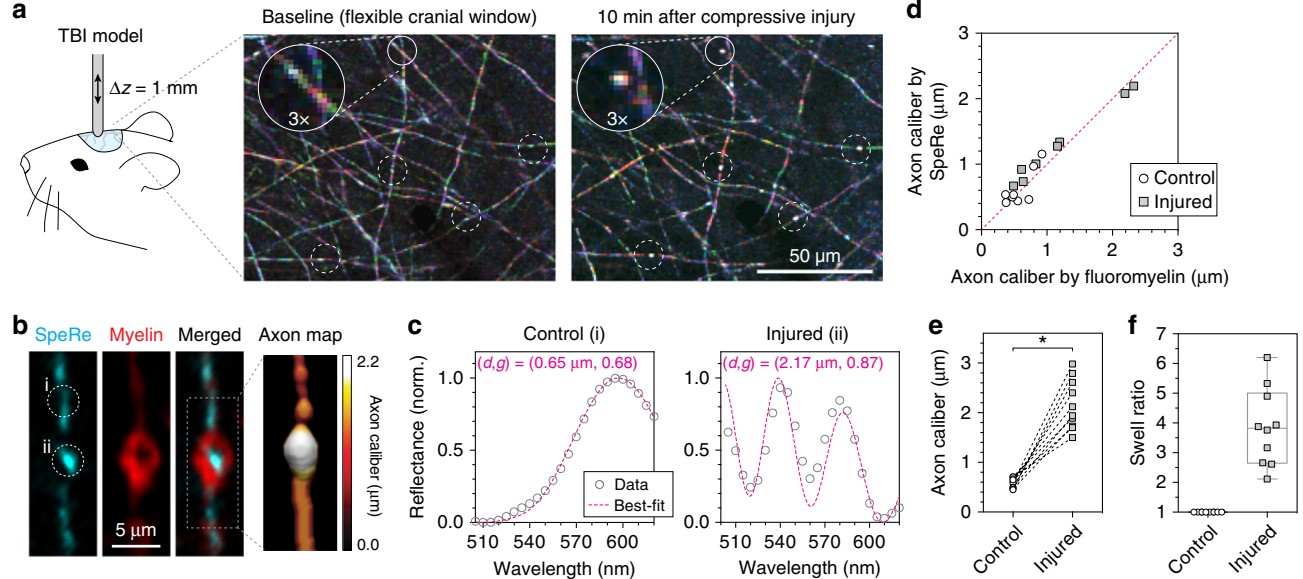

**Fig. 6** In vivo imaging of nanoscale axonal pathology in traumatic brain injury. **a** SpeRe images of the live mouse cortex (layer I) before and after mild compressive injury (1 mm indentation for 30 s). White circles indicate injured axon segments. **b** An injured axon from a cortical tissue slice imaged ex vivo using reflectance (cyan) and flouromyelin fluorescence (red). The axon map was reconstructed using the axon diameters obtained by fitting the measured spectrum for each segment to simulation data. **c** Reflectance spectra of the nearby control and injured axon segments, which are indicated as 'i' and 'ii', respectively, in **b**. Dotted lines in magenta are simulation curves that provide the best fit to experimental spectra (caliber = 0.65 μm, g-ratio = 0.68 for control; caliber = 2.17 μm, g-ratio = 0.87 for injured). **d** Validation of axon caliber measurement by SpeRe in injured axons. Dotted line in magenta is a unity line ($R^2 = 0.94$, $n = 10$ axons). **e** Quantification of axon diameter in nearby control and injured axon segments. **f** Swell ratio in the bulged axon quantified by dividing the axon diameter in the bulging region by the axon diameter in the nearby control segment. $*p < 0.0001$ (paired $t$-test, two-sided)

**Myelin swelling under osmotic challenge**. As living matter, myelin dynamically regulates its internal state in response to external physicochemical perturbations. For instance, a change in osmotic pressure induces volumetric changes in myelin, which occurs predominantly in the extracellular space[31, 32, 37]. In earlier simulations, we found that the sub-nanometer changes in the individual extracellular space are encoded as spectral shifts in visible reflectance (Fig. 2f, g). To first test whether SpeRe is robust enough to sense nanostructural changes, we performed repeated measurements on a synthetic plastic fiber and a spinal axon (Supplementary Fig. 10). In both samples, we obtained sub-nanometer precision in spectral peaks (standard deviation: ±0.39 nm for a plastic fiber and ±0.89 nm for a spinal axon, $n = 8$ for each sample). Having validated the robustness, we then performed SpeRe on a brain slice in the context of physiologic osmotic modulation (Supplementary Fig. 11). Using the fluidic system, the osmotic pressure of the medium varied from iso-smotic pressure (300 mOsm L$^{-1}$) to hypo- or hyper-osmotic pressures within the physiologic range, and SpeRe was performed on the same axon repeatedly. After the induction of osmotic swelling or shrinkage, myelinated axons exhibited a spectral shift in reflectance spectrum (Fig. 5a, b). Although there was variability in the degree of spectral shift among axons, higher osmotic pressure consistently led to a blue shift in the reflectance spectrum, which agrees with our simulation results (Fig. 2f, g). By contrast, we did not observe any significant structural changes using conventional confocal microscopic imaging on flouromyelin-stained samples (Supplementary Fig. 12). We next estimated the swelling ratio of the extracellular space from spectral shift using the linear relationship revealed in the simulation. Following an increase in the osmolarity to 360 mOsm L$^{-1}$, the spectral shift was $-8.5 \pm 2.9$ nm, and following a decrease in the osmolarity to 240 mOsm L$^{-1}$, it was $15.8 \pm 5.3$ nm (Fig. 5c). According to the linear relationship (Fig. 2g) these spectral shifts were converted to a swelling ratio ($r_s$) of 0.82 and 1.33,

corresponding to 0.9 nm shrinkage and 1.7 nm swelling of each extracellular layer (5 nm at isosmotic pressure), respectively. This result demonstrates that SpeRe can detect nanoscale structural dynamics in myelin.

**In vivo application in traumatic brain injury**. Diffuse axonal injury is one of the most common pathological hallmarks of traumatic brain injury by mechanical insult[38]. To test whether nanostructural modifications of myelinated axons occur under mechanical stress, we performed SpeRe imaging on a live mouse before and after a mild compressive insult (Online Methods). For in vivo SpeRe imaging, we used a custom-designed upright confocal microscopy with an array spectral detector, providing fast and reliable spectral acquisition under physiologic motion (Supplementary Fig. 3). To access the brain cortex in vivo, we used a soft cranial window made of a flexible elastomeric window[39], which enabled repeated follow-up imaging along with compressive mechanical stimuli between imaging sessions. We applied pressure to compress the cortical surface by a magnitude of 1 mm over a period of 30 s. The mild indentation did not result in either detectable tissue deformation or injury to the neuronal cell body when examined in post-mortem Nissl stain (Supplementary Fig. 13). Nonetheless, SpeRe imaging on the cortex revealed remarkable acute changes in axons following mild compression (Fig. 6a). Notably, we observed the formation of dispersed breaks along with a highly reflective intervening spot for each break.

To understand the structural origin of these changes, we performed spectral analysis on the injured axon in vivo. Consistently, the reflective spot at the break had a reflectance spectrum of higher periodicity compared to that of nearby control regions (Fig. 6b, c). When converted to axon diameter by assuming that the myelin thickness is constant, these changes in spectral periodicity corresponded to the several-fold ($2.86 \pm 0.41$) enlargement of the axon, indicating bulging of the axon in the

ROI (Fig. 6d–f). Fluoromyelin counterstaining and a reconstructed axon map confirmed that the reflective spot was indeed bulging (Fig. 6b). This observation, that various insults on axons can induce the swelling by impaired macromolecular transport, is in agreement with previous studies[40].

Axon bulging involves enlargement of the axon diameter without any thickening of the myelin sheath, resulting in an increase in the g-ratio. For instance, if the axon bulges by a factor of 2, the g-ratio would increase from 0.7 to 0.82. To analyze the concomitant change in the axon diameter and the g-ratio, we used a multiparametric fitting that extracts the parameters from the simulated spectrum providing the best fit to a measured spectrum (online Methods). By applying this analysis to injured axons, we consistently observed the bulging-induced increase in the g-ratio, that is, enlargement of the axon caliber (Fig. 6c). We also found that estimation of the axon caliber becomes highly precise by calibrating the g-ratio to bulged axons (Supplementary Fig. 14). Collectively, these results suggest that SpeRe is broadly applicable to neuropathologic conditions involving multiple structural changes simultaneously.

## Discussion

We have reported a new imaging modality, termed SpeRe. SpeRe is developed based on the previously reported technique SCoRe, which pioneered in vivo application of optical reflectance for qualitative imaging of myelinated axons[22]. In this work, we first introduced spectroscopic analysis of broadband light reflection and obtained quantitative information at nanoscale. Our theoretical simulation clarified the physical principles of SpeRe and further revealed quantitative metrics, such as overall reflectance for the degree of myelination, wavenumber period for axon diameter, and spectral shift for myelin swelling. These metrics are demonstrated to be highly useful in neuroscience applications, as exemplified in studies on osmotic swelling and traumatic brain injury. Notably, our new tool revealed the prevalence of axon bulging in the context of mild compressive injury and measured physiologic swelling/shrinkage of the myelin sublayer with a nanometer accuracy. By permitting unprecedented access to nanostructural information in living milieu, SpeRe opens a new scope of opportunities in unveiling nanoscale dynamic phenomena on myelinated axons in health and disease.

There are several limitations to this technique. First, it is based on light reflection; thus, in principle, it has a limited range of angles at which a signal is detectable. This angle was measured to be approximately ±10° around the image plane (Supplementary Fig. 6). This feature can introduce significant sample selection bias if not taken into consideration. Sample slicing or mounting in accordance with a priori geometric information, which is commonly used for axon tracing studies for a specific neural circuit, offers a solution. Additional angular scanning by tilting the specimen mount or rotating the objective nosepiece also ameliorates the issue of limited sampling. In the case of transparent samples, such as zebrafish embryos, tomographic rotation may be employed for comprehensive three-dimensional profiling[41]. Second, the interpretation of the spectral features is dependent on specific geometric models. The myelinated axon has a well-defined structure under physiologic conditions with a g-ratio of ~0.7, depending on anatomical regions. Although narrowly tuned in most anatomical regions, this uncertainty in the g-ratio can still compromise the measurement precision. For example, axons in the anterior commissure have a g-ratio of 0.72–0.79[42]. According to our simulation, this level of uncertainty compromises the measurement precision of axon diameter by up to ±4% (e.g., ±20 nm for $d = 500$ nm). This error can be avoided or minimized if we have a prior knowledge on the geometric model. In our study on traumatic brain injury, we observed that the axon was swollen by several-fold, and that considering this information led to more precise measurement of axon diameter. Thus, we recommend to validate a geometric model by other techniques (such as electron microscopy), especially for applications involving significant structural change, such as demyelinating diseases or the early development of myelin[22]. For a validated geometric model, the multiparametric fitting approach is feasible for acquiring multiple structural parameters of interest simultaneously by solving the inverse problem (Fig. 6c). Third, light-based techniques intrinsically suffer from limited penetration into biological tissues. In the brain cortex, the $1/e$ attenuation length is measured to be ~20 μm, which is comparable to confocal fluorescence imaging (Supplementary Fig. 5). Existing solutions proposed for other optical modalities, such as longer wavelength[43], wave-front shaping[44], and endomicroscopy[45], would ameliorate the penetration issue.

The resolution of SpeRe is affected by the total spectral bandwidth of the light source and the spectral resolution of the detection. In our inverted setup with a bandwidth of 200 nm and a spectral resolution of 5 nm, axon diameters were measured within an accuracy of ~10 nm, assuming that our geometric assumption is valid. Our study using osmotic challenge demonstrated that it is possible to measure myelin swelling at the nanometer scale. More sensitive measurements can be achieved by improving the spectral linewidth and adapting the broadband source[46]. Remarkably, nanoscopic resolution is attained without introducing high light dose. In our experiments, we irradiated the specimen within 3–5 μW μm$^{-2}$ with a pixel dwell time of 5–10 μs. This fluence level (<0.05 J cm$^{-2}$) is approximately an order-of-magnitude lower than conventional confocal fluorescence imaging and several orders-of-magnitude less than super-resolution techniques. This dose is safely below the maximum permissive exposure level of ~1 J cm$^{-2}$ for human skin. Additionally, SpeRe does not involve any exogenous chemical or genetic labeling. Therefore, we expect the rapid adoption of this technique for studies of in vivo neurophysiology and myelin-associated diseases, as well as for clinical practices in diagnosis of demyelination and axonal injury. Integration with other complementary label-free imaging modalities is also desired. For example, coherent Raman microscopy can capture direct structural information of normal and diseased myelinated axons at submicron resolution[47, 48]. When combined with SpeRe, it would provide a refined geometric model, resulting in more precise quantification. Moreover, SpeRe is generally applicable to various transparent materials with cylindrical or spherical geometries, such as microfibers, microgels, and lipid-membrane vesicles (Supplementary Fig. 8). These synthetic structures may be developed as contrast agents for SpeRe.

## Methods

**Mice**. All mice were housed with littermates in groups of two to five in reverse day/night cycle and given ad libitum access to food and water. Male or female C57BL6J wild-type mice aged 7–12 weeks old (The Jackson Laboratory) were used for all studies. All animal experiments were performed in compliance with institutional guidelines and approved by the sub-committee on research animal care at Sungkyunkwan University.

**Preparation of nervous tissues**. Mice were fully anesthetized using 3% isoflourane (Hanapharm) and were subsequently decapitated rostral to the first cervical vertebra. The brain and the spinal cord were rapidly extracted and immersed in a cold dissection medium (~4 °C). The dissection medium was composed of 4 mM KCl, 128 mM NaCl, 21 mM NaHCO$_3$, 0.5 mM NaH$_2$PO$_4$, 30 mM glucose, 1 mM CaCl$_2$, and 1 mM MgSO$_4$ in distilled water, aerated with 95% O$_2$ and 5% CO$_2$[1]. The isolated brain was sliced into horizontal sections at a thickness of 200 μm using a vibratome (VT1200S, Leica). For the spinal cord, spinal nerves branching out at the lumbar spine were dissected. For myelin staining, tissues were immersed in a 1% dilution of a stock solution of fluoromyelin in PBS for 30 min

and washed 3 times with dissection medium. The tissue was then mounted on a confocal dish with a #1.5 glass bottom, gently compressed by a 5 mm round coverslip with its rim glued to the dish with a light-cured acrylic resin, and immersed in cold recording solution prepared by adding 1 mM $CaCl_2$ to the dissection medium[49]. All experimental procedures, including sample preparation and imaging, were completed within 2 h to avoid potential sample degradation.

**Soft cranial window model.** For SpeRe imaging with mechanical perturbation, a flexible cranial window model was used as described previously[39]. Briefly, each mouse was fully anesthetized with 2.0% of isoflurane, the scalp was incised, and a metal chamber (Narishige) was affixed to the skull using a light-cured acrylic resin (OA2, Dentkist). An approximately 3 mm diameter portion of the cranium was gently removed with a microdrill, with care taken to prevent damaging the cortex. The exposed cortex was then sealed by a transparent, flexible PDMS (poly-dimethylsiloxane) film by gluing the film into the skull with cyanoacrylate glue. The exposed skull around the film was covered with a light-cured acrylic resin to form a well to hold water for the dipping lens.

**Optics setup.** To perform SpeRe in slice preparations, we used an inverted confocal microscopy body (Leica SP8) coupled to a broadband supercontinuum laser (NKT photonics). The input laser was spectrally scanned across the visible wavelength (470–670 nm) at 5 nm intervals by AOTF (acousto-optical tunable filter). A resonant 2-axis galvanomirror (12 kHz) and a piezoelectric stage were used for $xy$ and $z$ scanning, respectively. The scanned beam was focused through a water-immersion objective lens (20X, NA 0.75). The sample was mounted on a piezo-electric $z$-stage. The reflected beam from the sample was directed to the detector by AOBS (acousto-optical beam splitter), spatially filtered with a confocal pinhole (1 Airy unit), and detected using a photomultiplier tube (PMT). After spectral scanning, fluoromyelin fluorescence images were acquired with excitation at 560 nm and emission at $650 \pm 25$ nm. For in vivo imaging, we used a custom-designed upright system (Bruker Intravital) coupled to a broadband supercontinuum laser (NKT photonics) and a spectrometer (Andor). Reflectance images and spectrums were acquired with a water immersion apochromatic objective lens (20X NA 1.0).

**Data acquisition and processing.** The image stack acquired by spectral scanning (470–670 nm at 5 nm intervals) was normalized by input laser intensity measured by a mirror sample and registered by cross-correlation if translation or rotation was noted (stackreg, ImageJ). The reflectance spectrum for each pixel was extracted in the region of interest (ROI) for further analysis. For robust quantitative analysis, acquired data with low signal-to-background ratio (SBR < 3) were excluded. In our study on osmotic challenge, we introduced spatial averaging to improve the signal-to-noise ratio. We manually set the ROI along the axon (contour length ~ 5 μm) in a reflectance image (Supplementary Fig. 15). In order to avoid partial-volume artifact, the ROI for each spectrum was carefully chosen to have structural homogeneity (i.e., spectral homogeneity), which was later confirmed by the goodness-of-fit ($R^2 > 0.7$) during a multiparametric analysis.

For quantification of the axon diameter, the spectrum was plotted in wavenumber domain, and its mode functions were decomposed via EMD. Spectral trends that produced extensive noise, which affects the reflectance spectrum, were removed. From the multiple modes, typically ranging from 1 to 6, we selected a mode function that had the highest correlation with the original spectrum. The period of the mode function was then determined as the peak interval of spectrum for each pixel. Each period was converted to axon diameter using the formula obtained from the wave simulation (Fig. 2e). For quantification of swelling ratio, the spectrum was low-pass filtered, and the spectral shift of the maxima was quantified and converted to a swelling ratio as per the formula obtained from the wave simulation (Fig. 2g).

For a multiparametric analysis, we obtained a large-scale simulation dataset at a range of axon diameters and $g$-ratios (i.e., lookup table). To reduce computational burden, we limited the range of axon diameters near to the pre-estimated diameter by spectral periodicity. For each measured spectrum, we selected the best-fit simulated spectrum based on Pearson correlation ($R^2 > 0.7$), and extracted the axon diameter and the $g$-ratio (Fig. 6b, c). We confirmed that the empirical threshold of '$R^2 > 0.7$' effectively filtered out artifactual spectrums (e.g., partial-volume artifact, movement artifact). The extracted axon diameters for each axon segment were linearly interpolated and reconstructed in three-dimensional view using a rendering software (Imaris, Bitplane).

**Electron microscopy.** Tissue samples for electron microscopy were prepared as described previously[50]. Briefly, small pieces of brain and spinal cord tissues (~1 $mm^3$) were acquired from the freshly euthanized mice. Using the Epon812 protocol, samples were fixed, stained, and sliced into sections that were 70 nm thick. The samples were mounted on a grid and imaged using transmission electron microscopy (Tecnai G2).

For imaging monodisperse polystyrene microbeads with a nominal diameter of 10 μm (72822, Sigma-Aldrich), the stock solution was dried on a cover glass, gold-coated using a physical vapor deposition, and imaged using a scanning electron microscopy (JEOL JSM-700M).

**Traumatic brain injury.** To mimic traumatic brain injury, we applied a mild compressive injury to the cortex through a flexible elastomeric window. A rod (1 mm in diameter) with a round tip was mounted on the objective turret with a servo motor so that the center of the imaging field of view was targeted simply by rotating the turret. After taking baseline images, the turret was rotated, and compressive injury was applied (1 mm indentation from the dura for 30 s). The follow-up images were obtained after the injury, and mice were euthanized for post-mortem histological analysis.

**Osmotic challenge.** Brain slices were mounted on a fluidic chamber filled with CSF at an isosmotic pressure of 300 mOsm $L^{-1}$. Hypotonic (270, 240 mOsm $L^{-1}$) and hypertonic (330, 360 mOsm $L^{-1}$) aCSF were prepared by decreasing the concentration of NaCl (hypotonic) or adding mannitol (hypertonic), respectively[51]. The bathing solution was changed by infusion through a syringe. After changing the bathing solution, samples were incubated for at least 15 min to achieve osmotic equilibria.

**Histology.** After cardiac perfusion, the brain from each mouse was extracted and fixed in 4% paraformaldehyde for 24 h. The fixed tissue was cryoprotected with a 30% sucrose solution in saline for 2 days. Each brain was frozen-sectioned into 40 μm thick sections before undergoing Nissl staining. Sections were imaged with a bright-field microscope (Leica) with a 10X objective lens (NA 0.3).

**Statistical analysis.** We used GraphPad Prism for statistical analyses. Paired or unpaired $t$-tests (two-sided) were used for group comparisons by assuming normality based on previous literatures. Neither randomization nor blinding was applied. For regression analyses, we presented correlation coefficient ($R^2$) along with the sample size. The data presented are either as mean ± standard error or box-and-whisker plots. We considered a $p$-value less than 0.05 to be statistically significant.

**Code availability.** The Matlab codes used for the simulation studies are available from the corresponding author upon reasonable request for purposes of academic research.

**Data availability.** All relevant data are available from the corresponding author upon reasonable request.

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

## Acknowledgements

This work was supported by the Institute for Basic Science (IBS-R015-D1) and the National Research Foundation of Korea (2016R1A6A3A11936389). We are grateful to Seonghoon Kim, Woei-Ming Lee, Sheldon J.J. Kwok, and Jun-kyo Francis Suh for their constructive discussions.

## Author contributions

M.C. initiated and supervised the study. J.K. performed the experiments and data analysis. M.K. performed optical simulations. B.K., Y.J., and O.J. contributed to image processing and data quantification. H.P. and J.H.K. contributed to animal surgery, in vivo imaging, and post-mortem histology. S.H.Y., M.S., and S.K. contributed to data interpretation and editing of the manuscript. M.C. and J.K. wrote the manuscript with input from all authors.
