## [Peer Review File · Nature Communications]

Reviewers' comments:

Reviewer #1 (Remarks to the Author):

Kwon et al. describe the use of spectral reflectometry (SpeRe), a label-free optical technique, to study the myelinated axons in tissue and in vivo. The work is overall interesting and could be a powerful technique for looking at, in details, how myelinated axons change in structure. The work could be suitable for publication in Nature Communications if the author could satisfactorily address the points below:

1. Overall, one major problem with the manuscript is that it has not been put into good context with previous studies on spectral confocal reflectance microscopy (SCoRe). Although the SpeRe method developed in the current manuscript is a major advance to allow for quantification of axon diameters, etc., the underlying principles of spectral reflectance are still the same with SCoRe. The fact that SCoRe first enabled high-contrast, labeling-free imaging of myelinated axons is certainly the foundations of this work. Previous SCoRe work only used 3 wavelengths and the results were qualitative. In this work by having the full spectral information the authors were able to quantify the results. These points need to be made clearer in the revision: this reviewer considers that would still make the current manuscript suitable for Nature Communications.
2. The title is very unclear –this would make readers think that this work is about super-resolution microscopy of cultured neurons. Should emphasize below points in the title: label-free, quantitative examination, myelinated axons, and in tissue / in vivo.
3. For super-resolution, the references cited (19,20) are not very relevant as they are mostly with cultured cells on coverslips. Two other references by the same labs would be more appropriate: (Science 339, 452; this showed results of tissues in the supplement), and (PNAS 114, E191; this paper focuses on myelination).
4. For Fig. 2bd, the authors should consider drawing the absolute values of reflectance (fraction of the total amount of incident light), instead of the normalized values, which could be more helpful.
5. For Fig. 2de, it would be helpful to also include the corresponding wavelength (in nm) as secondary axes on the top (d) and on the right (e), in addition to the primary axes in wavenumber, for easier comparison with other discussion in text.
6. "Another notable feature was the localization of the reflectance signal only at the centerline, which was clearly apparent in relatively large spinal axons": this was already discussed in Ref 18. Should discuss that here.
7. From Fig. 5b, it appears that for the 283 mOSm/L sample, in addition to a redshift, there is also an increase in wavelength period –does this mean a change in axon diameter? Overall, can changes in axon diameter be fully decoupled from myelin swelling?

8. For Fig. 6, it appears bulges are randomly generated along the axons. What do the bulges actually correspond to, and do these results suggest that the rest parts (non-bulge) of the axons did not change in structure? To help visualize the changes, it would be helpful to plot Fig. 6a as the mapped local diameter as opposed to the raw data as presented now.

9. Page number of Ref 31 is incorrect.

Reviewer #2 (Remarks to the Author):

In this manuscript Kwon et al., adapt and validate spectral reflectometry (SpeRe), as a novel methodology for imaging live axons in the CNS. Using this technology the authors demonstrate the feasibility to measure structural changes in thinly myelinated CNS axons (up to 6 wraps) at the nanoscale level. The data is convincing and beautifully presented. The authors demonstrated that SpeRe signal is produced by myelinated axons and is very sensitive to changes in myelin thickness up to 5-6 wraps. They were also able to precisely measure the changes in axon diameter resulting from the characteristic molecular subdomains of myelinated axons (nodes vs. internodes and non-compact myelin) as well as those resulting from changes in osmolarity and following trauma. Their findings are properly discussed in the context of existing techniques, and they acknowledged both the advantages and disadvantages of this approach. The main limitation as recognized by the authors is the "sample bias" of this method towards the population of small diameter, thinly myelinated axons. However, to my knowledge, none of the optical technologies available to image live axons can sample this population or even thicker axons at this level of resolution. I anticipate that this method might be especially useful for imaging of remyelinated axons, which display thinner myelin after recovery compared to uninjured axons. Currently, there is no label-free method, other than electron microscopy, that allows the identification of these axons. Overall, I'm very enthusiastic about this work and highly recommend its publication.

Reviewer #3 (Remarks to the Author):

The manuscript uses a technique called SPeRe to determine the size of axon diameters and to some extent myelin thickness in fixed and live tissue. Essentially, the technique measures the spectrum of the confocal reflectance images and extracts subresolution information from the shape of the spectrum. The SPeRe technique is a close cousin of the SCoRe technique published by Schain in Nat Met (2014) but is different: the SCoRe technique put forward confocal reflectance as a method to measure myelination in general, but did not extract detailed information about the myelin thickness. In this respect, the present article is novel and distinct.

The authors show a detailed modelling of the SPeRe contrast in cylindrical structures, which is warranted by the fact that the SPeRe technique itself is not new, but it is used with flat structures in the semiconductor industry, as acknowledged by the authors. The models may

be unnecessarily complex for what is needed, since they consider diffraction but only the central part of the myelin ring is yielding a signal, which may be sufficiently modelled with a plane wave model. The conclusion of the modelling is that axon caliber can be determined from the modulation. However, the inverse relationship determined from the model depends on the g-ratio, which itself depends on myelin thickness, and that thickness is not measured directly (below, they assume $g = 0.7$). I would argue when they say that it "only modestly contributed to this inverse relationship." that it affects it more than what they claim.

- COMMENT TO ADDRESS: This incomplete two-variable relationship modelled with a simple inverse law is a weakness of the technique and should be emphasized more. The lack of knowledge of the myelin thickness leads to an uncertainty in the axon caliber that can be significant (from figure 2e, loss like $\pm 10-20\%$ for low caliber axons). It should be discussed.

- COMMENT TO ADDRESS: The blue shifting of the spectrum with myelin thickness may not be as robust as described here. The description in the caption is not clear either: "f, Simulated reflectance spectra with a varying swelling ratio of the extracellular space between the myelin layers. The axon size and g-ratio are set to $0.5 \mu\text{m}$ and 0.7 , respectively.". If the g ratio and the axon diameter are both set, then the swelling ratio is always 1.0. Obviously, something is amiss here: what is changed in the model? The myelin thickness, therefore the g ratio is "approximately 0.7".

The experiments on nervous tissue are interesting and convincing. It would be appropriate to credit Schain et al more, as many images obtained here are also available in that publication. The biggest weakness in the present work compared to Schain, is that as they push the technique further to extract myelin thickness, they actually lack a good method to measure the said myelin. The fluoromyelin labelling shown here only labels the outer rim, which is not the best control. I find that the section starting at: "To validate the source of the signal,..." until the end of the paragraph does not teach us more than we know from Schain et al.

- COMMENT TO ADDRESS: I suggest to shorten that paragraph or indicate "As shown in Schain et al, ..."
- COMMENT TO ADDRESS: I don't find the Schmitt Latnterman incisure images insightful as they are presented. It is not clear exactly what can be extracted from the images. 3d and 3e indicate that the incisures and the paranoids seem to have similar spectral signatures. Please discuss more how each was identified and provide convincing evidence (exogenous labelling would be ideal).

The section on axon caliber mapping is well done, assuming the g ratio does not change.

- COMMENT TO ADDRESS: The section on myelin swelling is also interesting but figure 5b looks very different from the other graphs shown elsewhere (such as 2f, which should be similar). Explain the very different curves (no clear modulation with possibly large

background).

- COMMENT TO ADDRESS: The comment "According to the linear relationship shown in Fig. 2g, these spectral shifts were converted to a swelling ratio (r_s) of 0.82 and 1.33, corresponding to 0.9 nm shrinkage and 1.7 nm swelling for each extracellular layer in myelin, respectively." makes little sense to me: if the swelling ratio of 0.82 and 1.33 correspond to changes of 0.9 nm and 1.7 nm, then it means the normal diameter is approximately 1 nm, which makes no sense. I can only assume this is a typo and they meant 0.9 micrometers and 1.33 micrometer. If this is not the case, please explain thoroughly.

The section on traumatic injury describes in detail an application of degeneration in live animals. This follows well from the previous experiments, but the sentence: "When converted to axon diameter," is not clear. How exactly is the fluoromyelin used to estimate axon caliber? It appears that the myelin thickness is assumed constant (which is certainly correct), but it is not stated until the following paragraph.

- COMMENT TO ADDRESS: Please discuss exactly how the reconstructed axon caliber map was obtained in Figure 6. If a constant myelin is assumed, please mention it. And exactly what thickness is assumed? How is it determined? It seems to me the sentence "g ratio from 0.7 to 0.84" should really be "0.7 to 0.85" if the caliber is doubling.

- COMMENT TO ADDRESS: It seems to me the sentence: "By applying this analysis to injured axons, we consistently observed the bulging-induced increase in g-ratio as well as enlargement of the axon caliber (Fig. 6c)." is redundant: if the axon caliber increases, the g ratio will increase assuming a constant thickness. Rewrite that sentence.

IN the discussion, everything is clear and well laid out: it is labels-free, low power. The resolution depends on spectral bandwidth. This is appropriate and well discussed. I would say the biggest weakness of the technique is that myelin thickness is not measured directly which is a big assumption for the present technique,. I am surprised the work in coherent Raman microscopy (Pubmed: CARS and myelin) is not referenced. This is not an easy technique to integrate, but it should be discussed as it has proven to be quite sensitive in measuring myelin thickness.

OVERALL: I find this is a very nice article with an extension to a technique (SCoRe) yielding more information. The weakness is the indirect myelin thickness measurement. I think it could be published after addressing my comments.

Response to Reviewers

We thank the editor and reviewers for their constructive comments and suggestions. We have provided a detailed point-by-point response below. All changes in the revised manuscript are highlighted in yellow.

Reviewer #1

Kwon et al. describe the use of spectral reflectometry (SpeRe), a label-free optical technique, to study the myelinated axons in tissue and in vivo. The work is overall interesting and could be a powerful technique for looking at, in details, how myelinated axons change in structure. The work could be suitable for publication in Nature Communications if the author could satisfactorily address the points below:

Reply: We thank the reviewer for the constructive comments. We have addressed all the comments as detailed below.

1. Overall, one major problem with the manuscript is that it has not been put into good context with previous studies on spectral confocal reflectance microscopy (SCoRe). Although the SpeRe method developed in the current manuscript is a major advance to allow for quantification of axon diameters, etc., the underlying principles of spectral reflectance are still the same with SCoRe. The fact that SCoRe first enabled high-contrast, labeling-free imaging of myelinated axons is certainly the foundations of this work. Previous SCoRe work only used 3 wavelengths and the results were qualitative. In this work by having the full spectral information the authors were able to quantify the results. These points need to be made clearer in the revision: this reviewer considers that would still make the current manuscript suitable for Nature Communications.

Reply: We fully agree with the reviewer that SpeRe is certainly grounded on the previous SCoRe technique (Schain AJ et al., *Nature Medicine* 2014)¹ and that our original manuscript has not been clear on this technical history. To clarify this, we have revised the last paragraph of the Introduction (p3, line 28-34). Additionally, we have elaborated contribution of the SCoRe work to our findings (please see our responses to comment 6 by this reviewer and comment 5 by the reviewer 3).

- Page 3: “Recently, Schain et al. showed that spectral reflectance from nervous tissues provides label-free, high contrast imaging of myelinated axons in vivo²⁰. This technique, named SCoRe, utilized several discrete laser-lines for generating reflectance images of myelinated axons and was shown to be highly useful for qualitative detection of myelin development and degeneration. Here, we report that spectroscopic analysis of broadband light reflection from myelinated axons provides quantitative examination of the multilayered cytoarchitecture at nanoscale precision.”

2. The title is very unclear –this would make readers think that this work is about super-resolution microscopy of cultured neurons. Should emphasize below points in the title: label-free, quantitative examination, myelinated axons, and in tissue / in vivo.

Reply: We appreciate the reviewer for suggesting a clearer title. To include the points suggested by the reviewer, we have edited the title to “Label-free nanoscale optical metrology on myelinated axons *in vivo*” (p1, line 1).

3. For super-resolution, the references cited (19,20) are not very relevant as they are mostly with cultured cells on coverslips. Two other references by the same labs would be more appropriate: (Science 339, 452; this showed results of tissues in the supplement), and (PNAS 114, E191; this paper focuses on myelination).

Reply: We thank the reviewer for recommending more suitable references. We have replaced the references as suggested (p3, line 27).

- References

18. Xu, K., Zhong, G. & Zhuang, X. *Actin, Spectrin, and Associated Proteins Form a Periodic Cytoskeletal Structure in Axons. Science 339, 452–457 (2013).*

19. D’Este, E., Kamin, D., Balzarotti, F. & Hell, S. W. *Ultrastructural anatomy of nodes of Ranvier in the peripheral nervous system as revealed by STED microscopy. Proc. Natl. Acad. Sci. 114, E191–E199 (2017).*

4. For Fig. 2bd, the authors should consider drawing the absolute values of reflectance (fraction of the total amount of incident light), instead of the normalized values, which could be more helpful.

Reply: Our original description might have not been clear. In Fig. 2b, 2d, and 2f, the y-axes indicate absolute reflectance (%). To clarify this, we have included the explanation on how we obtain optical reflectance in the figure legend (p19, figure caption for Fig. 2).

- Fig. 2: “Absolute reflectance (%) for each wavelength is simulated by applying the theory of electromagnetic waves.”

5. For Fig. 2de, it would be helpful to also include the corresponding wavelength (in nm) as secondary axes on the top (d) and on the right (e), in addition to the primary axes in wavenumber, for easier comparison with other discussion in text.

Reply: As suggested, we have added the corresponding wavelength axes in Fig. 2d. However, Fig. 2e was kept unchanged because there is no absolute one-to-one relationship between wavenumber period and wavelength interval (p19, Fig. 2d).

Revised figure 2d

6. “Another notable feature was the localization of the reflectance signal only at the centerline, which was clearly apparent in relatively large spinal axons”: this was already discussed in Ref 18. Should discuss that here.

Reply: As suggested, we have clarified that this observation was already discussed in the SCoRe paper (p5, lines 25-27)¹. Please also refer our response to comment 5 by the reviewer 3.

- Page 5: “As shown in Schain et al., the reflectance signal was specifically observed only at myelinated segments and was located only at the centerline”

7. From Fig. 5b, it appears that for the 283 mOSm/L sample, in addition to a redshift, there is also an increase in wavelength period –does this mean a change in axon diameter? Overall, can changes in axon diameter be fully decoupled from myelin swelling?

Reply: In our study on osmotic challenge, we assumed that the major structural change should occur at the extracellular layers of myelin, which was supported by previous studies. Blaurock et al. (*Brain Research*, 1981)² studied changes in subcellular structures in myelinated axons using X-ray diffraction and observed that only the extracellular space in the myelin is predominantly modulated. Additionally, Benoit et al. (*Neuroscience*, 1996)³ induced axon swelling by treating Ciguatoxin (CTX), which irreversibly opens voltage-dependent sodium channels. The authors observed that only the nodal regions were swollen and internode showed no apparent change, indicating that the myelinated portion of the axon is resilient to osmotic swelling. Collectively, these studies suggest that osmotic modulation induces structural change predominantly in the extracellular space, without significantly affecting the axon diameter. Indeed, our dataset (n = 4) exhibited apparent change in spectral shift, without notable change in spectral periodicity (i.e. axon size). To further validate if wavenumber periodicity is unchanged, we performed Fourier analysis for the original spectrums in Fig. 5b (please see the Figure for Reviewer 1). The Fourier-domain data did not show any notable change in the peak position. To conclude, our assumption on decoupling between axon diameter and myelin sheath (i.e. extracellular space) is valid in our experimental context.

Figure for Reviewer 1 | Fourier transform of the spectrums in Fig. 5b.

Yet, we agree to the reviewers' points that the original dataset presented in Fig. 5b is noisy and could be perceived differently (please also see our reply to the comment 7 by the reviewer 3). To quantitatively investigate this, we compared the original dataset with the simulation. The original dataset showed good fit to the simulation, but also contained noise (Figure for Reviewer 4). Therefore, we have changed the dataset in Fig. 5b (p22, revised Fig. 5b) to a new dataset having larger axon diameter, which clearly represents spectral shift by osmotic modulation.

Revised Fig. 5b

8. For Fig. 6, it appears bulges are randomly generated along the axons. What do the bulges actually correspond to, and do these results suggest that the rest parts (non-bulge) of the axons did not change in structure? To help visualize the changes, it would be helpful to plot Fig. 6a as the mapped local diameter as opposed to the raw data as presented now.

Reply:

“What do the bulges actually correspond to”

Axon bulging is one of well-known features of axonal degeneration observed in various neurodegenerative disorders (Yang Yi et al., *Trends in Neurosciences* 2013)⁴. Cellular mechanism of axon bulging in response to focal injury has been studied recently. In brief, the bulges correspond to accumulation of autophagosome-like vacuoles and aggregated proteins. The accumulation is triggered

by impaired axonal transport in regions of collapsed cytoskeleton. We have elaborated this cellular mechanism in the revised manuscript (p8, lines 15-16).

- Page 8: *“This observation is in agreement with previous studies, that various insults on axons can induce the swelling by impaired macromolecular transport³⁹.”*

“do these results suggest that the rest parts (non-bulge) of the axons did not change in structure? To help visualize the changes, it would be helpful to plot Fig. 6a as the mapped local diameter as opposed to the raw data as presented now.”

We agree with the reviewer that it would be helpful to compare the nanostructure of the axons before and after the mechanical injury. This experiment will directly address if non-bulged axons are structurally intact. However, we found that this experiment is technically challenging with our current experimental setting. Even with metal fixatives attached to the cranium, there is significant physiologic motion. To minimize the motion-induced artifact, our imaging protocol for *in vivo* studies was optimized to rapid line-scan on a small segment (~20 um) along the axon. As the axon bulging is sparse, the probability that we get the paired observation of uninjured and injured axon is quite low. To mitigate this problem, we are developing more stable fixatives and optimized scanning scheme for our future study.

9. Page number of Ref 31 is incorrect.

Reply: We have corrected the page number (p15, line 32).

Reviewer #2

In this manuscript Kwon et al., adapt and validate spectral reflectometry (SpeRe), as a novel methodology for imaging live axons in the CNS. Using this technology the authors demonstrate the feasibility to measure structural changes in thinly myelinated CNS axons (up to 6 wraps) at the nanoscale level. The data is convincing and beautifully presented. The authors demonstrated that SpeRe signal is produced by myelinated axons and is very sensitive to changes in myelin thickness up to 5-6 wraps. They were also able to precisely measure the changes in axon diameter resulting from the characteristic molecular subdomains of myelinated axons (nodes vs. internodes and non-compact myelin) as well as those resulting from changes in osmolarity and following trauma. Their findings are properly discussed in the context of existing techniques, and they acknowledged both the advantages and disadvantages of this approach. The main limitation as recognized by the authors is the “sample bias” of this method towards the population of small diameter, thinly myelinated axons. However, to my knowledge, none of the optical technologies available to image live axons can sample this population or even thicker axons at this level of resolution. I anticipate that this method might be especially useful for imaging of remyelinated axons, which display thinner myelin after recovery compared to uninjured axons. Currently, there is no label-free method, other than electron microscopy, that allows the identification of these axons. Overall, I’m very enthusiastic about this work and highly recommend its publication.

Reply: We appreciate the reviewer for recommending publication.

Reviewer #3

The manuscript uses a technique called SPeRe to determine the size of axon diameters and to some extent myelin thickness in fixed and live tissue. Essentially, the technique measures the spectrum of the confocal reflectance images and extracts subresolution information from the shape of the spectrum. The SPeRe technique is a close cousin of the SCoRe technique published by Schain in Nat Met (2014) but is different: the SCoRe technique put forward confocal reflectance as a method to measure myelination in general, but did not extract detailed information about the myelin thickness. In this respect, the present article is novel and distinct.

Reply: We appreciate the reviewer for insightful comments and suggestions.

1. The authors show a detailed modelling of the SPeRe contrast in cylindrical structures, which is warranted by the fact that the SPeRe technique itself is not new, but it is used with flat structures in the semiconductor industry, as acknowledged by the authors. The models may be unnecessarily complex for what is needed, since they consider diffraction but only the central part of the myelin ring is yielding a signal, which may be sufficiently modelled with a plane wave model.

Reply: To rigorously estimate the reflectance spectrum, our optical simulation was performed based on the theory of electromagnetic (EM) waves as described in Supplementary Information. The vector diffraction theory is introduced which accounts for polarization of the incident light at the focus for every incident angle. This precise theoretical model is chosen considering our high-NA (> 0.7) optical system and small axons (typically, 300–1000 nm). The most simplified model would be modeling axons as a planar substrate (i.e. thin-film) and incident beam as a plane wave. To test if the plane-wave model suffices, we performed optical simulations on the same axons using both EM-wave model and plane-wave model. The resulting spectra showed similar features but are significantly different (Figure for Reviewer 2).

Figure for Reviewer 2 | Comparison between the EM-wave model and plane-wave model.

2. The conclusion of the modelling is that axon caliber can be determined from the modulation. However, the inverse relationship determined from the model depends on the g-ratio, which itself depends on myelin thickness, and that thickness is not measured directly (below, they assume $g = 0.7$). I would argue when they say that it "only modestly contributed to this inverse relationship." that it affects it more than what they claim. This incomplete two-variable relationship modelled with a simple inverse law is a weakness of the technique and should be emphasized more. The lack of knowledge of the myelin thickness leads to an uncertainty in the axon caliber that can be significant (from figure 2e, loss like $\pm 10\text{-}20\%$ for low caliber axons). It should be discussed.

Reply: We appreciate the reviewer for pointing out the critical point. As the reviewer commented, the g-ratio affects the inverse relationship. However, the distribution of the g-ratio in physiologic condition is revealed to be narrowly tuned, for example 0.72–0.79 for the anterior commissure⁵. At this physiological variability in g-ratio, measurement precision of SpeRe on axon diameter is about $\pm 4\%$, irrespective of the axon diameter (Figure for Reviewer 3). For a typical axon size of 500 nm, the measurement error is up to ± 20 nm. We have elaborated this limitation in the discussion section (p9, lines 15-26).

- Page 9: *“The myelinated axon has a well-defined structure under physiologic conditions with a g-ratio of ~ 0.7 , depending on anatomical regions. Although narrowly-tuned in most anatomical regions, this uncertainty in the g-ratio can still compromise the measurement precision. For example, axons in the anterior commissure has the g-ratio of 0.72–0.79³⁴. According to our simulation, this level of uncertainty compromises the measurement precision of axon diameter by up to $\pm 4\%$ (e.g. ± 20 nm for $d = 500$ nm). This error can be avoided or minimized if we have a prior knowledge on the geometric model. In our study on traumatic brain injury, we observed that the axon was swollen by several-fold, and considering this information led to more precise measurement of axon diameter. Thus, we recommend to validate a geometric model by other techniques (such as, electron microscopy), especially for applications involving significant structural change, such as demyelinating diseases or the early development of myelin²⁰.”*

Figure for Reviewer 3 | The measurement error of SpeRe by the variation of the g-ratio.

3. The blue shifting of the spectrum with myelin thickness may not be as robust as described here.

Reply: In principle, SpeRe is robust because it measures self-interference from the cavity formed by the multilayered cellular structure, which is analogous to Fabry-Perot etalon. The detection of sub-nanometer spectral shift by self-interference is well-established in the field of whispering-gallery mode sensors. Although *in vivo* imaging in the scattering media poses a technical challenge, we think our study on osmotic challenge clearly demonstrate the proof-of-principle of SpeRe in detecting nanoscale structural change.

4. The description in the caption is not clear either: "f, Simulated reflectance spectra with a varying swelling ratio of the extracellular space between the myelin layers. The axon size and g-ratio are set to 0.5 μm and 0.7, respectively.". If the g ratio and the axon diameter are both set, then the swelling ratio is always 1.0. Obviously, something is amiss here: what is changed in the model? The myelin thickness, therefore the g ratio is "approximately 0.7".

Reply: We thank the reviewer for pointing out the error in our description. As the reviewer commented, the myelin thickness indeed changes, even though the change is at nanoscale and does not significantly affect the g-ratio. We assumed that only the extracellular space in the myelin sheath is changed based on previous studies (ref. 30, 33). As the extracellular space occupies $\sim 28\%$ (e.g. 5 nm out of 18 nm in each myelin layer) of the myelin sheath, the swelling ratio of 1.2 led to the thickening of each extracellular space by 1 nm and that corresponds to $\sim 5\%$ increase in the myelin sheath (each myelin layer increases from 18 to 19 nm). In case of the swelling ratio of 1.2, the g-ratio decreases from 0.7 to 0.69. To avoid the confusion, we have edited the manuscript as follows (p19, figure caption for Fig. 2f):

- Page 19: "*The axon size and g-ratio at the isosmotic condition are set to 0.5 μm and 0.7 (i.e. 6 myelin layers), respectively.*"

5. The experiments on nervous tissue are interesting and convincing. It would be appropriate to credit Schain et al more, as many images obtained here are also available in that publication. The biggest weakness in the present work compared to Schain, is that as they push the technique further to extract myelin thickness, they actually lack a good method to measure the said myelin. The fluoromyelin labelling shown here only labels the outer rim, which is not the best control. I find that the section starting at: "To validate the source of the signal,..." until the end of the paragraph does not teach us more than we know from Schain et al. I suggest to shorten that paragraph or indicate "As shown in Schain et al, ..."

Reply: We fully agree with the reviewer that our original manuscript has not been clear on the previous contribution by Schain AJ et al. (*Nature Medicine* 2014)¹. We have edited throughout the manuscript to credit the previous publication more (please see our response to the comment 1 by the reviewer 1). In addition, we have edited the section on source validation as suggested (p5, lines 24-27).

- Page 5: "*To validate the source of the signal, we counterstained with a fluorescent probe*

selective to myelin (fluoromyelin)³². As shown in Schain et al., the reflectance signal was specifically observed only at myelinated segments and was located only at the centerline (Fig. 3a)²⁰.”

6. I don't find the Schmitt Latnterman incisure images insightful as they are presented. It is not clear exactly what can be extracted from the images. 3d and 3e indicate that the incisures and the paranoids seem to have similar spectral signatures. Please discuss more how each was identified and provide convincing evidence (exogenous labelling would be ideal).

Reply: Our approach to identify the incisure and the node (or paranode) is grounded on the previous report by Schain et al¹. In the SCoRe paper, the authors showed that spectral reflectance with several laser-lines can be used to identify specialized axonal structures, such as Schmidt-Lantermann incisure (SLi) and node of Ranvier. These structural features were validated by mT/mG transgenic mice, which expresses tdTomato in cell membranes. As our method acquires optical reflectance at full visible range, we could also detect the same reflectance features. In a representative SpeRe images, SLi can be easily located by their characteristic feature (Figure for Reviewer 4).

Figure for Reviewer 4 | Schmidt-Lantermann incisure has a comb-like structure in SpeRe.

As the incisure and the paranode had specialized subcellular structures, we expected that these features should be reflected in the reflectance spectra. We indeed observed specialized spectral features as shown in Fig. 3d,e. We have added detailed description on these spectral features as follows (p6, lines 1-15):

- Page 6: *“It has also been reported that spectral reflectance can be used to identify specialized structures in the axon, such as Schmidt-Lantermann incisure and node of Ranvier²⁰ (Fig. 3c). We further studied if SpeRe on these specialized structures reveals more structural information³³. To visualize spectral information along the longitude of the axon, we presented the data as a spectral map ($x\lambda$). In the spectral map, the incisure, a cone-shaped loosening of the myelin sheath, showed a characteristic speckled pattern (Fig. 3d, lower panel). This speckled spectral feature is conceivably due to abrupt longitudinal change at the loosening of each myelin layer.”*
- Page 6: *“In the spectral map, the paranode often showed a progressive spectral shift, conceivably due to the gradual structural change. These observations indicate that spectral features may offer a way to quantify geometric parameters of the axon, such as the length of the incisure, node of Ranvier, and internode, in a label-free manner.”*

7. The section on axon caliber mapping is well done, assuming the g ratio does not change. The section on myelin swelling is also interesting but figure 5b looks very different from the other graphs shown elsewhere (such as 2f, which should be similar). Explain the very different curves (no clear modulation with possibly large background).

Reply: To confirm that the measured spectrum in the original Fig. 5b is consistent with our simulation, we searched for simulation data showing the best-fit to the measured spectrum (Figure for Reviewer 5). The measured spectrum exhibited high similarity to the simulation data with axon diameter (d) of 510 nm and g -ratio of 0.7. Yet, we noted that the measured spectrum is noisy and does not visualize spectral shift clearly, presumably due to background contamination (note that there is unexpected shoulder at ~ 560 nm).

Figure for Reviewer 5 | Measured spectrum at isosmotic condition in the original Fig. 5b (open circles) merged with a best-fit simulation data ($d = 510$ nm, g -ratio = 0.7).

As larger axons are more suitable for clearly visualize the spectral shift, we have changed the representative dataset to another dataset having the axon diameter of ~ 770 nm (revised Fig. 5b). We also confirmed that the new dataset shows excellent fit to the simulation.

Revised Fig. 5b

8. The comment "According to the linear relationship shown in Fig. 2g, these spectral shifts were converted to a swelling ratio (r_s) of 0.82 and 1.33, corresponding to 0.9 nm shrinkage and 1.7 nm swelling for each extracellular layer in myelin, respectively." makes little sense to me: if the swelling

ratio of 0.82 and 1.33 correspond to changes of 0.9 nm and 1.7 nm, then it means the normal diameter is approximately 1 nm, which makes no sense. I can only assume this is a typo and they meant 0.9 micrometers and 1.33 micrometer. If this is not the case, please explain thoroughly.

Reply: We apologize to the reviewer for the confusion. The subcellular region affected by osmotic change is mainly the extracellular space^{2,3}, thus the swelling ratio is calculated by considering only the extracellular space. In our geometric model, thickness of the extracellular space is 5 nm for each myelin layer. In case of the swelling ratio of 0.82, the extracellular space shrinks by 18%, therefore the change in thickness for each extracellular space is

$$5 \text{ nm} \times 18\% = 0.9 \text{ nm}$$

Similarly, the swelling ratio of 1.33 corresponds to

$$5 \text{ nm} \times 33\% \approx 1.7 \text{ nm}$$

To clarify this, we have revised the manuscript as follows (p7, lines 23-26).

- Page 7: *“According to the linear relationship shown in Fig. 2g, these spectral shifts were converted to a swelling ratio (r_s) of 0.82 and 1.33, corresponding 0.9 nm shrinkage and 1.7 nm swelling of each extracellular layer (5 nm at isosmotic pressure), respectively.”*

9. The section on traumatic injury describes in detail an application of degeneration in live animals. This follows well from the previous experiments, but the sentence: "When converted to axon diameter, ..." is not clear. How exactly is the fluoromyelin used to estimate axon caliber? It appears that the myelin thickness is assumed constant (which is certainly correct), but it is not stated until the following paragraph.

Reply: As the reviewer mentioned, we assumed that the myelin thickness is constant to estimate the axon caliber. We have elaborated this information in the revised manuscript (p8, lines 10-13).

- Page 8: *“When converted to axon diameter by assuming that the myelin thickness is constant, these changes in spectral periodicity corresponded to the several-fold (2.86 ± 0.41) enlargement of the axon, indicating bulging of the axon in the region of interest (Fig. 6d-f).”*

10. Please discuss exactly how the reconstructed axon caliber map was obtained in Figure 6. If a constant myelin is assumed, please mention it. And exactly what thickness is assumed? How is it determined? It seems to me the sentence "g ratio from 0.7 to 0.84" should really be "0.7 to 0.85" if the caliber is doubling.

Reply: As the reviewer pointed out, we assumed a constant g-ratio of 0.7. As the g-ratio is defined as the ratio of the inner axonal diameter (d) to the total outer diameter (D), the g-ratio can be expressed as follows.

$$g - \text{ratio} = \frac{d}{D} = 0.7$$

Accordingly, myelin thickness can be expressed as follows.

$$\text{myelin thickness} = D - d = \frac{d}{0.7} - d = \frac{3}{7}d$$

In our model, the myelin thickness has a discrete value determined by the number of myelin layer (N). According to our geometric model, the myelin thickness can be expressed as

$$\text{myelin thickness} = p + N(m + c + m + e) - e$$

where p , pericellular space = 12 nm; m , membrane = 5 nm; c , cytosol = 3 nm; e , extracellular space = 5 nm; N is an integer. For example, if $d = 1 \mu\text{m}$, myelin thickness is ~ 428 nm. The number of myelin layer (N) is determined as follows.

$$N = \text{round} \left\{ \frac{428 - p + e}{2m + c + e} \right\} = \text{round} \left\{ \frac{428 - 12 + 5}{18} \right\} = 23 \text{ layers}$$

We have elaborated this information in the Supplementary Information (p2, section 1.2).

It seems to me the sentence "g ratio from 0.7 to 0.84" should really be "0.7 to 0.85" if the caliber is doubling.

If a myelinated axon having a g-ratio of 0.7 is swollen by factor 2, the g-ratio increases to 0.82, which can be calculated as follows.

$$g - \text{ratio} = \frac{2d}{2d + 3/7d} = \frac{2}{17/7} = 0.82$$

We have corrected the error in the revised manuscript (p8, line 19).

11. It seems to me the sentence: "By applying this analysis to injured axons, we consistently observed the bulging-induced increase in g-ratio as well as enlargement of the axon caliber (Fig. 6c)." is redundant: if the axon caliber increases, the g ratio will increase assuming a constant thickness. Rewrite that sentence.

Reply: We have revised the sentence as suggested (p8, 22-23).

- Page 8: "*By applying this analysis to injured axons, we consistently observed the bulging-induced increase in the g-ratio, that is, enlargement of the axon caliber (Fig. 6c).*"

12. In the discussion, everything is clear and well laid out: it is labels-free, low power. The resolution depends on spectral bandwidth. This is appropriate and well discussed. I would say the biggest weakness of the technique is that myelin thickness is not measured directly which is a big assumption for the present technique. I am surprised the work in coherent Raman microscopy (Pubmed: CARS and myelin) is not referenced. This is not an easy technique to integrate, but it should be discussed as it has proven to be quite sensitive in measuring myelin thickness.

Reply: We agree that information on the myelin would significantly improve the precision of this approach. We hope our reply to the comment 2 satisfactorily addressed the reviewer's point.

I am surprised the work in coherent Raman microscopy (Pubmed: CARS and myelin) is not referenced. This is not an easy technique to integrate, but it should be discussed as it has proven to be quite sensitive in measuring myelin thickness.

We agree with the reviewer that the multimodal imaging system integrating SpeRe and CARS would be highly beneficial. Because both modalities are label-free, we think it would be readily translated into the clinic. We have elaborated this idea in the Discussion section (p10, lines 11-13).

13. OVERALL: I find this is a very nice article with an extension to a technique (SCoRe) yielding more information. The weakness is the indirect myelin thickness measurement. I think it could be published after addressing my comments.

Reply: We appreciate the reviewer for detailed and insightful comments. We believe our replies and revised manuscript satisfactorily addressed the concerns and suggestions raised by the reviewer.

Additional change

- In abstract, results of the current study are written in present tense.
- The Methods section is moved to the main text.
- Units are changed according to the guideline.
- To clarify the statistical methods, we included inclusion/exclusion criteria and the information on randomization/blinding (p12-13, Online Methods)
- In the Acknowledgements, the grant number has been updated (p17).

References

1. Schain, A. J., Hill, R. A. & Grutzendler, J. Label-free in vivo imaging of myelinated axons in health and disease with spectral confocal reflectance microscopy. *Nat. Med.* **20**, 443–449 (2014).
2. Blaurock, A. E. The spaces between membrane bilayers within PNS myelin as characterized by X-ray diffraction. *Brain Res.* **210**, 383–387 (1981).
3. Benoit, E., Juzans, P., Legrand, A. & Molgo, J. Nodal swelling produced by ciguatoxin-induced selective activation of sodium channels in myelinated nerve fibers. *Neuroscience* **71**, 1121–1131 (1996).
4. Yang, Y., Coleman, M., Zhang, L., Zheng, X. & Yue, Z. Autophagy in axonal and dendritic degeneration. *Trends Neurosci.* **36**, 418–428 (2013).
5. Chomiak, T. & Hu, B. What is the optimal value of the g-ratio for myelinated fibers in the rat CNS? A theoretical approach. *PLoS One* **4**, e7754 (2009).

Reviewers' comments:

Reviewer #1 (Remarks to the Author):

The authors did a great job in the revision. I recommend publication.

Reviewer #3 (Remarks to the Author):

Many concerns have been addressed.

However, I don't find the response fully satisfying. For instance, Rev #3, comment 3 simply says that they think it's robust, but I still don't think they have shown robustness with their work. If this is in Nature Communications, it will need to be better than that: people will try to use the technique for their own experiment. It needs to be more solidly demonstrated in my opinion.

I also don't find that just adding a reference to Schain or is sufficient to their their work in the context of everything that has been done in the field of myelin characterization. Coherent Raman (by Cheng, Cote and others) has also been used extensively to characterize myelin, and the present manuscript only lightly mentions it with a single reference.

Also, comment 7, reviewer 3: I again don't find the "presumably due to background contamination" satisfying: this is a significant bump that appears like a modulation, and the technique is all about characterizing these modulations.

Overall, I like the technique, but this is not fully worked out to a Nature Communication Level article. I don't think others can take this technique with confidence into their lab.

Response to Reviewer

We thank the reviewers for their insightful comments. We have provided a detailed point-by-point response below. All changes in the revised manuscript are highlighted in yellow.

Reviewer #1

1. The authors did a great job in the revision. I recommend publication.

Reply: We thank the reviewer for recommending publication.

Reviewer #3

1. Many concerns have been addressed. However, I don't find the response fully satisfying. For instance, Rev #3, comment 3 simply says that they think it's robust, but I still don't think they have shown robustness with their work. If this is in Nature Communications, it will need to be better than that: people will try to use the technique for their own experiment. It needs to be more solidly demonstrated in my opinion.

Reply: We fully agree that the robustness of this technique should be solidly validated. To verify the robustness of SpeRe, we performed the repeated measurements on a synthetic polymeric fiber and a spinal axon over the duration of ~25 min (**Supplementary Fig. S10**). We observed that the SpeRe measurement is indeed highly stable. The spectral shift measured by the peak position was less than ± 1 nm. The variability in standard deviation ($n = 8$) was ± 0.39 nm for the synthetic fiber and ± 0.89 nm for the myelinated axon. This subnanometer-scale noise is about an order-of-magnitude smaller than the spectral shifts observed in our most sensitive study on osmotic modulation (Fig. 5b; spectral shift: ~10 nm). Therefore, the contribution of the measurement noise to our studies is negligible.

- Page 7: *“To first test whether SpeRe is robust enough to sense nanostructural changes, we performed repeated measurements on a synthetic plastic fiber and a spinal axon (Supplementary Fig. S10). In both samples, we obtained subnanometer precision in spectral peaks (standard deviation: ± 0.39 nm for a plastic fiber and ± 0.89 nm for a spinal axon, $n = 8$ for each sample). Having validated the robustness, we then performed SpeRe on a brain slice in the context of physiologic osmotic modulation (Supplementary Fig. S11).”*

Figure S10 | Robustness of SpeRe. **a**, Repeated SpeRe measurements on a synthetic PMMA fiber and a spinal axon over time ($n = 8$ each). **b**, Reflectance spectrums over time. **c**, Reflectance spectrums from the dotted wavelength regions in (a). **d-e**, Quantification of spectral shifts (standard deviation: ± 0.39 nm for the PMMA fiber, ± 0.89 nm for the axon).

2. I also don't find that just adding a reference to Schain or is sufficient to their their work in the context of everything that has been done in the field of myelin characterization. Coherent Raman (by Cheng, Cote and others) has also been used extensively to characterize myelin, and the present manuscript only lightly mentions it with a single reference.

Reply: We appreciate that the myelin field has been significantly advanced by the development of various imaging tools. Especially coherent Raman microscopy can provide detailed structural information on the myelin in label-free and demonstrated to be applicable to demyelinated states. Therefore, integration with SpeRe would be highly advantageous by providing *a priori* knowledge on a geometric model for more precise quantification. We have elaborated the manuscript to clarify these points and also cited the relevant papers as suggested (Wang H et al., Biophysical Journal, 2005; Bélanger, E. et al. Journal of Biomedical Optics, 2011).

- Page 3: “Various contrast mechanisms for myelinated axons have been developed,

including fluorescence with exogenous probes^{13,14}, optical coherence tomography¹⁵, Raman scattering^{16,17}, and third-harmonic generation¹⁸. These technical advances enabled observation of dynamic cellular processes in live myelinated axons in physiology and pathophysiology. In particular, label-free techniques, such as Raman scattering and third-harmonic generation, have high potential for clinical translation^{18,19}. However, these microscopic techniques have spatial resolutions greater than the optical diffraction-limit of ~200 nm, rendering them inadequate for studying the nanostructures of interest in the myelinated axon (e.g. a cytosolic layer in myelin ≈ 3 nm).”

- Page 6: *“It has also been reported that multiple label-free techniques, including spectral reflectance²², coherent Raman scattering¹⁷, and third-harmonic generation¹⁸, can be used to identify specialized axonal structures, such as Schmidt-Lantermann incisure and node of Ranvier (Fig. 3c).”*
- Page 10: *“Integration with other complementary label-free imaging modalities is also desired. For example, coherent Raman microscopy can capture direct structural information of normal and diseased myelinated axons at submicron resolution^{47,48}. When combined with SpeRe, it would provide a refined geometric model, resulting in more precise quantification.”*

Additionally, we have re-emphasized the contribution of the prior work by Schain et al. in the discussion.

- Page 9: *“We have reported a new imaging modality, termed SpeRe. SpeRe is developed based on the previously reported technique, SCoRe, which pioneered in vivo application of optical reflectance for qualitative imaging of myelinated axons²². In this work, we first introduced spectroscopic analysis of broadband light reflection and obtained quantitative information at nanoscale. Our theoretical simulation clarified the physical principles of SpeRe...”*

3. Also, comment 7, reviewer 3: I again don't find the "presumably due to background contamination" satisfying: this is a significant bump that appears like a modulation, and the technique is all about characterizing these modulations.

Reply: We apologize that our previous response was insufficient. As shown in the map of axon caliber, the diameter is not uniform even along a single fiber (Fig. 4b). Thus, if an axon segment

contains heterogeneous structures (e.g. axon diameter), the resulting spectrum can be complicated by linear summation of multiple spectra. This so-called “partial-volume artifact” is a common issue in biomedical imaging. Indeed, the bump-like feature that we observed in the previous version of Fig. 5b was originated from summation of two spectra with distinct periodicity (**Figure for Reviewer 1**). To avoid this artifact, we confirmed that the axon segment in the current Fig. 5b does not contain any heterogeneous spectral features. In addition, we included the region-of-interest used in our analysis (Fig. 5a) and clarified the Methods section.

- Page 12: “In order to avoid partial-volume artifact, the region-of-interest for each spectrum was carefully chosen to have structural homogeneity.”

Additional Changes

1. To demonstrate the superiority of SpeRe to the widely-used confocal fluorescent microscopy, we have performed a control study with flouromyelin-stained axons under osmotic challenge. Despite of apparent macroscopic tissue-level swelling/deswelling, the diffraction-limited resolution (>200 nm) provided by confocal fluorescence imaging was insufficient to reliably capture the nanoscale structural changes (**Supplementary Fig. S12**).

- Page 7: “By contrast, we did not observe any significant structural changes using conventional confocal microscopic imaging on flouromyelin-stained samples (Supplementary Fig. S12).”

Figure S12 | Fluorescence-based quantification of myelin swelling. **a**, Confocal fluorescence images on flouromyelin-stained myelinated axons in a sliced cortical tissue under osmotic challenge. Scalebar, 3 μm. **b-c**, Quantification of the outer diameter and the inner diameter. Full-width-half-maximum (FWHM) was quantified from the intensity profile (n = 7 axons).

2. We validated that the SpeRe measurement is highly precise by using an electron microscopy, which is the current gold-standard technique (**Supplementary Fig. S9**).

- Page 6: “We further validated the SpeRe measurement on monodisperse synthetic microbeads by using a scanning electron microscopy (Supplementary Fig. S9).”
- Page 13: “For imaging monodisperse polystyrene microbeads with a nominal

diameter of 10 μm (72822, Sigma-Aldrich), the stock solution was dried on a cover glass, gold-coated using a physical vapor deposition, and imaged using a scanning electron microscopy (JEOL JSM-700M).”

Figure S9 | Precision of SpeRe. **a**, SpeRe on a monodisperse polystyrene bead with a nominal diameter of 10 μm . **b**, Scanning electron microscopy (SEM) on the monodisperse polystyrene beads. **c**, Comparison of diameters quantified by SpeRe (9.99 ± 0.07 , $n = 20$) and SEM (9.97 ± 0.10 , $n = 20$).

REVIEWERS' COMMENTS:

Reviewer #3 (Remarks to the Author):

The authors are doing a large amount of work, and I commend them for that. I think this is extremely interesting. However, there are two things that leave me hanging:

1) "Page 12: "In order to avoid partial-volume artifact, the region-of-interest for each spectrum was carefully chosen to have structural homogeneity."" This sounds like the ROIs is simply adjusted until "it works". The result probably depends on this ROI, which means the result of the future researchers who use this will also depend on the analysis. What is the strategy here to make this work an actual Method that other people will use?

2) The robustness is shown with phantoms and test systems, which is not what I think is required, although I find the SEM measurements very nice. However, this reflectance technique is used in the semiconductor industry, so we know this works. What we want to know is if it is robust for myelinated axons. People will want to invest time and money into making a similar system. It needs to be clear how the technique works and how to use it.

I will let the editor decide at this point.

Response to Reviewer

We thank the reviewer for insightful comments. We have provided a detailed point-by-point response below.

Reviewer #3

1. The authors are doing a large amount of work, and I commend them for that. I think this is extremely interesting. However, there are two things that leave me hanging: (1) "Page 12: "In order to avoid partial-volume artifact, the region-of-interest for each spectrum was carefully chosen to have structural homogeneity." This sounds like the ROIs is simply adjusted until "it works". The result probably depends on this ROI, which means the result of the future researchers who use this will also depend on the analysis. What is the strategy here to make this work an actual Method that other people will use?

Reply: We apologize that our previous description was unclear. We found that the partial-volume artifact creates the spectrum that significantly deviates from the simulated data. In our analysis based on the lookup table (i.e. large-scale simulation data), artifactual spectra were filtered out by the empirical threshold on goodness-of-fit ($R^2 > 0.7$). To clarify this, we have amended the Methods section and also included the Supplementary Fig. 15.

2. The robustness is shown with phantoms and test systems, which is not what I think is required, although I find the SEM measurements very nice. However, this reflectance technique is used in the semiconductor industry, so we know this works. What we want to know is if it is robust for myelinated axons. People will want to invest time and money into making a similar system. It needs to be clear how the technique works and how to use it. I will let the editor decide at this point.

Reply: We have shown a series of experimental validation on robustness, including a phantom study in parallel with SEM (Supplementary Fig. 10), and the measurement precision (Supplementary Fig. 10). As the reviewer commented, the principle of SpeRe should work on myelinated axons as long as our assumptions on physical parameters (refractive index and thickness) are valid. Ideally, this can be validated by direct comparison of SpeRe data with electron microscopy, but is experimentally highly challenging and burdensome. In the near future, we plan to adopt novel super-resolution techniques to clarify this issue.